# Dynamic Evaluation of Reward Models via Pairwise Maximum Discrepancy Competition

## Abstract

Reward models (RMs) are essential for aligning large language models with human preferences, making their rigorous and comprehensive evaluation a critical task. However, traditional evaluation methods rely heavily on closed datasets with pre-annotated preference pairs, which often fail to assess the generalization ability of RMs across unseen prompts in open-world scenarios. To overcome these limitations, we introduce the Pairwise Maximum Discrepancy Competition (PMDC) framework, a dynamic and annotation-efficient evaluation approach that adaptively selects informative test cases from a large, unlabeled, open-domain prompt pool. Specifically, the PMDC framework operates by first identifying input pairs that elicit significantly divergent preference scores from two RMs. These discriminative pairs are subsequently evaluated by an advanced large language model (LLM) acting as an oracle, determining which RM produces judgments more closely aligned with human preferences. The resulting pairwise comparisons are aggregated via the Bradley-Terry model, yielding an overall ordinal evaluation of the assessed RMs. We apply PMDC to re-evaluate 10 representative RMs from the RewardBench collection. The results reveal noticeable inconsistencies in RM rankings compared to those derived from conventional benchmarks. Further analysis uncovers the strengths and weaknesses of each model, providing valuable insights for future improvements in reward modeling.

## 1 Introduction

Reward models (RMs) are a cornerstone of modern alignment pipelines, enabling large language models (LLMs) to internalize complex human preferences through reinforcement learning from human feedback (RLHF) (Christiano et al., 2023; Stiennon et al., 2022). By learning to predict human preferences between pairs of model responses, RMs provide scalable training signals that guide LLMs toward desirable behaviors across various domains, including instruction following, reasoning, and safety (Bai et al., 2022; Ouyang et al., 2022; Feng et al., 2025). With the growing reliance on RMs for alignment, reliable, insightful, and scalable evaluation of RMs has become increasingly critical.

However, prevailing evaluation benchmarks (Lambert et al., 2024) for RMs predominantly rely on static, pre-annotated datasets that offer only a limited and often outdated representation of model capability. These conventional evaluation resources suffer from several critical limitations. First, their restricted coverage of the potential prompt and behavioral space impedes the assessment of model generalization to novel domains or edge-case scenarios. Second, the human annotations underpinning these datasets are typically sourced from specific demographic groups or constrained task contexts, potentially introducing biases that do not accurately reflect broader human judgment. Third, the fixed and publicly accessible nature of these test sets introduces inherent risks of overfitting, both explicit and implicit, where models may be optimized for benchmark performance without achieving meaningful improvements in alignment quality or robustness (Gao et al., 2022; Zhong et al., 2025; Kim et al., 2025).

To address these challenges, we propose the *Pairwise Maximum Discrepancy Competition* (PMDC), a dynamic and cost-efficient framework for evaluating RMs in open-world settings. Inspired by discrepancy-driven evaluation paradigms in computer vision (Saito et al., 2018), PMDC actively identifies prompt-response pairs that elicit highly divergent scores from pairs of reward models, so-

called maximum discrepancy samples. Such contentious instances are subsequently evaluated by a powerful LLM acting as an adjudicator, serving as a scalable proxy for human judgment. By aggregating these pairwise comparisons across a diverse pool of models and prompts, PMDC constructs a global ranking of RMs using the Bradley-Terry (BT) model (BRADLEY & TERRY, 1952), yielding both a holistic performance ranking and fine-grained insights into the relative strengths and weaknesses of each model.

Crucially, PMDC shifts the evaluation paradigm from static benchmarking to active and adaptive probing. Rather than assessing model performance on a fixed test set, it systematically identifies the most informative points of disagreement among RMs. This approach offers two key advantages. First, it enables *dynamic evaluation* by adaptively sampling test cases from an open-domain prompt pool and utilizing responses generated by a diverse set of LLMs, thereby facilitating the detection of out-of-distribution failures and enhancing generalization assessment. Second, it ensures *annotation efficiency* by submitting only the most discriminative sample pairs to the oracle for judgment, which significantly reduces annotation costs.

The main contributions of this work are threefold:

- **The first dynamic evaluation framework for reward models:** We introduce the PMDC framework, a novel evaluation paradigm that moves beyond static benchmarks by adaptively identifying high-discrepancy instances from an open-domain prompt pool.

- **A dynamically generated and actively probed evaluation dataset:** We construct a high-quality RM evaluation dataset via active probing to identify highly divergent reward model responses. Experiments demonstrate that it not only enables discrimination-rich evaluation of reward models but also enhances downstream alignment performance.

- **Empirical re-evaluation of state-of-the-art RMs with revealing insights:** We apply PMDC to 10 prominent reward models from the RewardBench collection (Malik et al., 2025), uncovering significant inconsistencies in their rankings compared to conventional benchmarks.

## 2 RELATED WORKS

### 2.1 REWARD MODEL BENCHMARKS

Early efforts in RMs evaluation, such as RewardBench (Lambert et al., 2024), primarily focused on measuring preference prediction accuracy within closed datasets. However, subsequent research has raised concerns about whether such narrow accuracy metrics reliably predict downstream alignment performance (Wen et al., 2025; LeVine et al., 2024). In response, more comprehensive benchmarks emerged, including RM-Bench (Liu et al., 2024b) and RewardBench 2 (Malik et al., 2025), which assess RMs on nuanced capabilities like discerning subtlety and resisting stylistic bias. This paradigm has further extended into specialized domains, reflecting the growing application scope of reward modeling. Recent benchmarks now assess multilingual (Gureja et al., 2025), vision-language (Yasunaga et al., 2025; Li et al., 2025), and embodied agent (Men et al., 2025) RMs. Concurrently, the adoption of powerful LLMs as reward models or preference judges has gained significant traction (Zheng et al., 2023; Dong et al., 2024). To formalize and standardize the assessment of these LLM-based evaluators, several dedicated benchmarks have been proposed (Thakur et al., 2025; Murugadoss et al., 2024; Tan et al., 2025; Zhou et al., 2025). These frameworks evaluate critical dimensions such as alignment with human preferences, robustness to varying instruction complexities, and consistency across diverse evaluation scenarios, establishing much-needed rigor in judge-style model assessment.

### 2.2 MAXIMUM DISCREPANCY COMPETITION

Beyond static benchmarks, it is crucial to actively and efficiently probe for model weaknesses. The Maximum Discrepancy (MAD) competition framework provides a powerful methodology for this task (Ma et al., 2020). Instead of relying on a fixed test set, MAD adaptively samples data points that cause the largest disagreement between two or more competing models. This principle has been successfully applied to expose failures and compare models in diverse domains, including objective

image quality (Ma et al., 2016) and semantic segmentation (Yan et al., 2021). More recently, this sample-efficient approach has been adapted for the human evaluation of large language models, demonstrating its effectiveness in identifying the most informative examples to distinguish between high-performing models (Feng et al., 2025). Our work is inspired by this adversarial, comparative approach to develop a more robust and efficient evaluation protocol for reward models.

## 3 PROPOSED PMDC

In this section, we introduce the PMDC framework to evaluate RMs. As illustrated in Figure 1, our method dynamically samples prompts from diverse datasets, generates responses using multiple LLMs, identifies maximum discrepancy pairs, obtains oracle judgments for contentious cases, and finally produces global rankings.

### 3.1 DATA GENERATION AND SCORING

The data generation begins with the construction of a diverse evaluation corpus through systematic sampling from multiple data sources. We first sample $M$ prompts from a comprehensive prompt pool comprising prompts aggregated from established benchmarks to ensure broad topical coverage. Concurrently, we sample $L$ LLM from a diverse model pool.

For each sampled prompt, we generate candidate responses using the selected LLMs, resulting in $M$ prompts each with $L$ potential responses. This process yields a dataset of prompt-response pairs that captures diverse response strategies and styles. The resulting collection forms our evaluation dataset $\mathcal{X} = \{(q_j, \{a_j^{(m)}\}_{m=1}^L)\}_{j=1}^M$.

Each prompt-response pair $(q, a)$ is then evaluated by $N$ distinct reward models, denoted $\mathcal{R} = \{R_i\}_{i=1}^N$. Each reward model $R_i$ assigns a real-valued score $s_i(q, a)$ reflecting its assessment of the response's quality. To enable a fair comparison across reward models that may operate on different numerical scales, we apply min-max normalization to the raw scores for each model $R_i$ across the entire dataset:

$$s_i'(q, a) = \frac{s_i(q, a) - \min_i}{\max_i - \min_i}, \tag{1}$$

where $\min_i$ and $\max_i$ represent the minimum and maximum scores produced by $R_i$ over the dataset, respectively. These normalized scores are converted into discrete preferences:

$$\text{Preference}(R_i, a_1, a_2) = \begin{cases} 1 & \text{if } s_i'(a_1) > s_i'(a_2) \\ 2 & \text{if } s_i'(a_1) < s_i'(a_2). \end{cases} \tag{2}$$

### 3.2 SAMPLE SELECTION

The objective is to efficiently evaluate and rank $N$ RMs on the dataset $\mathcal{X}$. Conventional evaluation methods heavily rely on static and pre-annotated datasets, which frequently fail to measure the generalization ability of RMs when confronted with unseen prompts in open-world settings. The standard process for evaluating RMs on new datasets consists of three stages. First, a small dataset $\mathcal{S}$ must be pre-selected. Second, predictions are generated by processing $\mathcal{S}$ through the RMs. Third, human evaluation is performed on these outputs to compare relative model performance. The RM that achieves the highest average subjective rating across $\mathcal{S}$ is considered superior. However, this evaluation paradigm is labor-intensive, expensive, and challenging to scale, posing significant practical limitations for efficient assessment of reward models.

Following the Maximum Discrepancy Competition (MAD) principle (Wang & Simoncelli, 2008), we aim to evaluate the RM by adaptively selecting a minimal yet highly informative subset of prompt-response pairs. We begin by considering the *simplest* scenario, where two RMs $R_A$ and $R_B$ are being compared under an oracle budget that permits the judgment of only one prompt and its corresponding response pair $(q, \{a_1, a_2\}) \in \mathcal{X}$. The core challenge thus reduces to: How can we automatically select the most informative sample from a large pool of candidates such that the relative performance between $R_A$ and $R_B$ can be most effectively discerned?

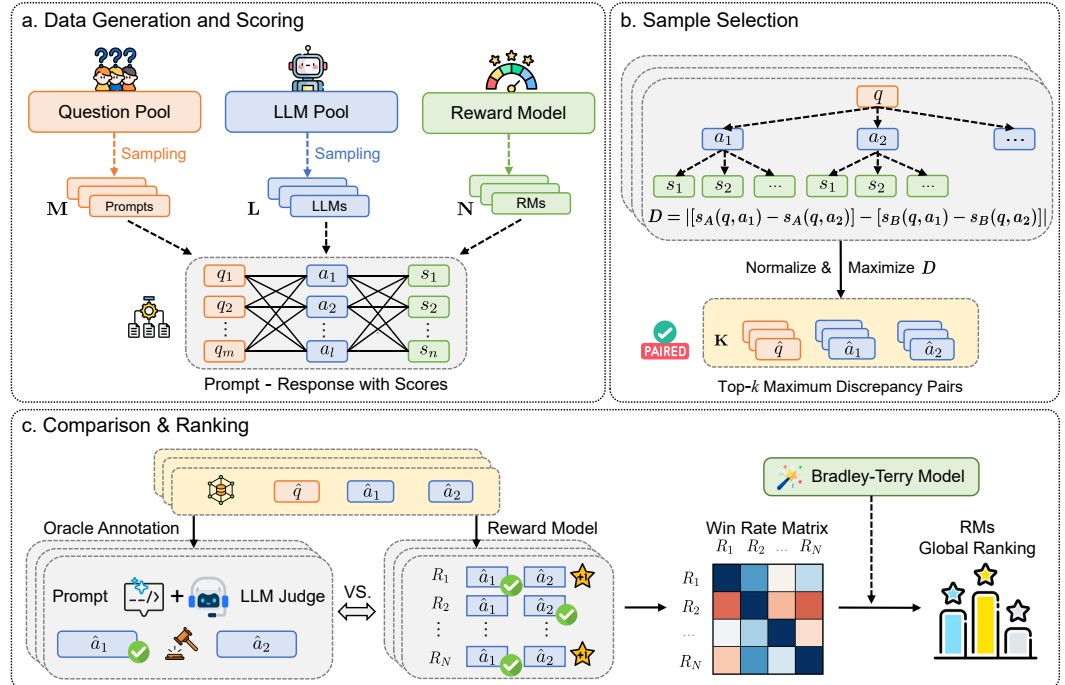

Figure 1: Overview of the proposed PMDC framework. (a) *Data generation and scoring*: Sample prompts and LLMs to build prompt-response pairs, which are then scored by reward models (RMs). (b) *Sample selection*: Based on the Maximum Discrepancy Competition principle, select top-$k$ pairs (with maximum RM preference discrepancy) to form an evaluation subset. (c) *Comparison & ranking*: Annotate the selected QA pairs with an Oracle (i.e., LLM-based Judge) to rank responses, compare Oracle results with RMs to build a win-rate matrix, and convert the pairwise comparisons into RMs' global ranking using the Bradley-Terry model.

According to the MAD competition methodology, PMDC selects the prompt-response pair $(\hat{q}, \{\hat{a}_1, \hat{a}_2\}) \in \mathcal{X}$ that best differentiates between RMs $R_A$ and $R_B$:

$$(\hat{q}, \{\hat{a}_1, \hat{a}_2\}) = \underset{(q, \{a_1, a_2\}) \in \mathcal{X}}{\arg\max} \left| [s_A(q, a_1) - s_A(q, a_2)] - [s_B(q, a_1) - s_B(q, a_2)] \right|, \qquad (3)$$

where $s_A(q, a_1) - s_A(q, a_2)$ represents the preference score difference assigned by model $R_A$ to the response pair $\{a_1, a_2\}$, with a larger positive value indicating a stronger preference for $a_1$ over $a_2$. The same applies to $s_B(q, a_1) - s_B(q, a_2)$ for $R_B$.

Then, we extend this idea to compare $R_A$ and $R_B$ over a small subset $\mathcal{S} \subset \mathcal{X}$ comprising $K$ prompt–response pairs with the highest discrepancy values, as computed by Eq. 3. The $k$-th pair is selected iteratively using:

$$(\hat{q}, \{\hat{a}_1, \hat{a}_2\})^{(k)} = \underset{(q, \{a_1, a_2\}) \in \mathcal{X} \backslash \mathcal{S}}{\arg\max} \left| [s_A(q, a_1) - s_A(q, a_2)] - [s_B(q, a_1) - s_B(q, a_2)] \right|, \qquad (4)$$

where $\mathcal{S} = \{(\hat{q}, \{\hat{a}_1, \hat{a}_2\})\}_{i=1}^{k-1}$ contains the previously chosen $k-1$ pairs. Each newly selected pair is incorporated into $\mathcal{S}$ for subsequent iterations.

### 3.3 COMPARISON & RANKING

The oracle assessment of the preferences from $R_A$ and $R_B$ for a given pair $(q, \{a_1, a_2\})$ leads two plausible results:

- The oracle's judgment is consistent with that of $R_A$ (or $R_B$). In this case, PMDC successfully identifies the most informative prompt–response pair for discriminating between the two models, thereby enabling a conclusive performance ranking.

- The oracle cannot determine a superior response, which is possible in open-world scenarios. Although the selected prompt-response pair $(\hat{q}, \{\hat{a}_1, \hat{a}_2\})$ may reveal divergent strengths (or weaknesses) of $R_A$ and $R_B$, but contributes less to their relative performance ranking.

Given $N$ RMs, PMDC chooses top-$k$ prompt-response pairs for each of the $\binom{N}{2}$ model pairs, resulting in a final evaluation set $\mathcal{D}$ of size $N(N-1)K$. Notably, the size of $\mathcal{D}$ is independent of the size of the input domain $\mathcal{X}$, allowing PMDC to benefit from an expanded $\mathcal{X}$ with broader prompt–response coverage.

For the Oracle assessment, PMDC employs a two-alternative forced choice (2AFC) paradigm. Each prompt–response pair $(q, \{a_1, a_2\}) \in \mathcal{S}$ is presented to the oracle alongside the outputs of two competing RMs, $R_A$ and $R_B$. The oracle is required to select the preferred response. The collected judgments are compiled into an $N \times N$ win-count matrix $W$, where $W_{i,j}$ records the number of votes for $R_i$ and against $R_j$. The symmetrized win rate matrix is computed as:

$$P_{i,j} = \frac{W_{i,j}}{W_{i,j} + W_{j,i} + \varepsilon}, \quad P_{i,i} = 0.5, \tag{5}$$

where $\varepsilon$ is a small smoothing constant, ensuring $P_{i,j} + P_{j,i} \approx 1$ off-diagonal and neutral diagonal.

We employ the BT model to infer the global ranking of $\mathcal{R}$. Specifically, we let $\xi$ be the vector of global ranking scores $[\xi_1, \ldots, \xi_n]$, and define the probability of $R_i$ being preferred over $R_j$ as

$$P_{i,j} = \frac{1}{1 + \exp(\xi_j - \xi_i)}. \tag{6}$$

We estimate the global scores by maximizing regularized log-likelihood with Broyden-Fletcher-Goldfarb-Shanno (BFGS) (Hunter, 2004), and applying L2 penalty ($\lambda = 10^{-6}$) for numerical stability and fixing $\xi_1 = 0$ for identifiability:

$$\log \mathcal{L}(\boldsymbol{\xi}) = \sum_{(i,j) \in \mathcal{C}} \left[ w_{ij} \log P_{i,j} + w_{ji} \log P_{i,j} \right] - \lambda \sum_{k=2}^{n} \xi_k^2. \tag{7}$$

We summarize the proposed PMDC in Algorithm 1.

## 4 EXPERIMENTS

In this section, we demonstrate the effectiveness of PMDC through comprehensive experiments on 10 prominent reward models. Section 4.1 details our setup, including datasets, reward models, and evaluation metrics. Section 4.2 presents PMDC's global rankings and compares them with established benchmarks. Section 4.3 analyzes PMDC's sensitivity to key design choices, such as top-$k$ selection, oracle judge, and sampling randomness. Finally, Section 4.4 demonstrates how PMDC-identified samples can improve reward models via targeted fine-tuning.

### 4.1 EXPERIMENTAL SETUP

**Dataset** We compile a large, unlabeled prompt pool by aggregating prompts from six established benchmarks to ensure broad topical coverage: 1) MMLU (Massive Multitask Language Understanding) (Hendrycks et al., 2021), 2) GSM8K (Grade School Math) (Cobbe et al., 2021), 3) HumanEval (Chen et al., 2021), 4) AlpacaEval (Li et al., 2023), 5) TruthfulQA (Lin et al., 2022), 6) HellaSwag (Zellers et al., 2019).

**Reward Models** We evaluate 10 representative RMs, covering different architectures, training paradigms, and parameter scales: 1) Skywork-Reward-Gemma-2-27B (Liu et al., 2024a), 2) QRM-Gemma-2-27B (Dorka, 2024), 3) Reward-Model-Mistral-7B-instruct-unified (Yang et al., 2024), 4) URM-LLaMa-3.1-8B (Lou et al., 2025), 5) ArmoRM-Llama3-8B-v0.1 (Wang et al., 2024), 6) Skywork-Reward-Llama-3.1-8B (Liu et al., 2024a), 7) Skywork-Reward-V2-Qwen3-8B (Liu et al., 2025), 8) Reward-Model-Deberta-v3-large-v2 (OpenAssistant, 2023), 9) Skywork-Reward-V2-Llama-3.2-3B (Liu et al., 2025), 10) Skywork-Reward-V2-Llama-3.1-8B (Liu et al., 2025).

---

**Algorithm 1** Pairwise Maximum Discrepancy Competition (PMDC)

---

1: **Input:** A Prompt pool, An LLM pool, Reward models $\mathcal{R} = \{R_1, \ldots, R_n\}$, Oracle $\mathcal{O}$, Top-$k$ parameter $k$.
2: **Initialize:** Pairwise win counts $W_{ij} = 0$ for all $i, j \in \{1, \ldots, n\}$.
3: Sample a batch of prompts $\mathcal{Q}$ from the Prompt pool
4: Sample a batch of response generators $\mathcal{G}$ from the LLM pool
5: **for** each prompt $q$ in $\mathcal{Q}$ **do**
6:     Generate response set $\{a_1, \ldots, a_m\}$ using generators $\mathcal{G}$.
7:     Compute all reward scores $s_i(q, a_j)$ for all $R_i \in \mathcal{R}$ and all $a_j$.
8:     Normalize scores: $s'_i(q, a_j) = \text{Min-Max}(s_i(q, a_j))$ for each model $R_i$.
9:     **Find Maximum Discrepancy samples:**
10:     Initialize discrepancy set $\text{MD}_{samples} = \emptyset$.
11:     **for** each model pair $(R_A, R_B) \in \binom{\mathcal{R}}{2}$ **do**
12:         Compute discrepancies $D_{A,B}(q, a_i, a_j)$ for all response pairs $(a_i, a_j)$ using Eq. 3.
13:         $\text{MD}_{\text{pair}} \leftarrow \text{top-}k\{(D_{A,B}, q, a_i, a_j, R_A, R_B) : \forall i < j\}$.
14:         $\text{MD}_{samples} \leftarrow \text{MD}_{samples} \cup \text{MD}_{\text{pair}}$.
15:     **end for**
16:     Total samples: $|\text{MD}_{samples}| = \binom{n}{2} \times k$.
17:     **Oracle Adjudication:**
18:     **for** each sample $(D, q, a_i, a_j, R_A, R_B)$ in $\text{MD}_{samples}$ **do**
19:         Get oracle preference $P_{\mathcal{O}} = \mathcal{O}(q, a_i, a_j)$.
20:         Get model preferences $P_A = (s'_A(a_i) > s'_A(a_j))$ and $P_B = (s'_B(a_i) > s'_B(a_j))$.
21:         **if** $P_A = P_{\mathcal{O}}$ and $P_B \neq P_{\mathcal{O}}$ **then**
22:             $W_{AB} \leftarrow W_{AB} + 1$.
23:         **else if** $P_B = P_{\mathcal{O}}$ and $P_A \neq P_{\mathcal{O}}$ **then**
24:             $W_{BA} \leftarrow W_{BA} + 1$.
25:         **end if**
26:     **end for**
27: **end for**
28: **Global Ranking:**
29: Compute win-rate matrix $P$ from $W$ using Eq. 5.
30: Estimate BT ranking score $\boldsymbol{\xi}$ by maximizing Eq. 7 using $W$.
31: **Output:** Global ranking based on $\boldsymbol{\xi}$.

---

Among them, models 1) and 2) are large-scale 27B parameter models that demonstrate superior performance, while models 3)-6) represent mid-scale instruction-tuned variants, and models 7)-10) are recent Skywork-V2 series models with various base architectures.

**Oracle** We employ Claude-Sonnet-4 as our oracle judge via the API endpoint. The oracle uses a systematic prompt design with structured instructions to ensure consistent and reliable judgments. The full prompt can be found in Appendix A1.

**Evaluation Metrics** In addition to the global ranking, we also employ the oracle agreement rate as an evaluation metric. The oracle agreement rate measures the proportion of Maximum Discrepancy samples where an RM's preference aligns with the oracle's judgment:

$$\text{Agreement}(R_i) = \frac{1}{|\mathcal{S}_i|} \sum_{s \in \mathcal{S}_i} \mathbb{I}\big[\text{Preference}(R_i, s) = \text{Oracle}(s)\big], \tag{8}$$

where $\mathcal{S}_i$ is the set of maximum discrepancy samples involving RM $R_i$. A higher value indicates that the RM is more reliable when evaluating challenging samples characterized by high inter-model disagreement.

**Implementation Details** For each experiment, we randomly select $1,000$ prompts from the compiled prompt pool. For each selected prompt, we generate 5 candidate responses by randomly sampling five LLMs from a diverse pool of twenty state-of-the-art models. For each reward model pair, we systematically select the top-$k$ QA pairs with the highest reward score discrepancy across all

Table 1: Global ranking results. Higher agreement indicates better oracle consistency on contentious Maximum Discrepancy samples.

| Model | Rank | BT ranking score | Agreement (%) |
|---|---|---|---|
| Skywork-Reward-Gemma-2-27B | 1 | 2.488 | 91.6 |
| QRM-Gemma-2-27B | 2 | 1.271 | 74.8 |
| Reward-Model-Mistral-7B-instruct-unified | 3 | 0.753 | 65.3 |
| URM-LLaMa-3.1-8B | 4 | 0.065 | 49.9 |
| ArmoRM-Llama3-8B-v0.1 | 5 | 0.000 | 48.3 |
| Skywork-Reward-Llama-3.1-8B | 6 | -0.185 | 44.1 |
| Skywork-Reward-V2-Qwen3-8B | 7 | -0.455 | 37.6 |
| Reward-Model-Deberta-v3-large-v2 | 8 | -0.507 | 36.7 |
| Skywork-Reward-V2-Llama-3.2-3B | 9 | -1.007 | 26.0 |
| Skywork-Reward-V2-Llama-3.1-8B | 10 | -1.059 | 24.9 |

$1,000 \times 5 = 5,000$ QA pairs using our normalized score difference metric (Eq. 3), where $k$ is a configurable hyperparameter (default $k = 10$). With 10 RMs, this yields $\binom{10}{2} \times k = 45 \times 10 = 450$ Maximum Discrepancy samples in total.

## 4.2 MAIN RESULTS

**Global Ranking Results** Table 1 presents the global ranking of the evaluated models, reporting both their BT ranking score and oracle agreement rates. As expected, higher BT scores correspond to greater consistency with the oracle's judgments, confirming the reliability of the estimated preferences. The pairwise win-rate landscape in Figure 2 provides a more detailed view of model performance. The results reveal a clear performance hierarchy with Skywork-Reward-Gemma-2-27B emerging as the top performer, achieving an oracle agreement rate of $91.6\%$ and the highest BT ranking score of $2.488$. This indicates consistently strong alignment with human-like judgments across contentious evaluation scenarios. QRM-Gemma-2-27B and Reward-Model-Mistral-7B-instruct-unified follow in second and third place, with agreement rates of $74.8\%$ and $65.3\%$, respectively. The dominance of Gemma-2-27B-based models in the top rankings suggests that scale and architecture significantly influence reward modeling capability. Oracle agreement rates complement the BT ranking by measuring absolute reliability on contentious samples, with the strong positive correlation confirming the robustness of our evaluation framework.

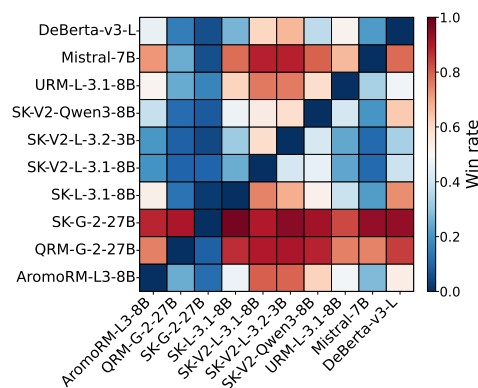

Figure 2: Pairwise win-rate heatmap across RMs on Maximum Discrepancy samples.

**Comparison Against Established Benchmarks** To validate the reliability of PMDC, we compare our rankings against RewardBench2, a challenging held-out evaluation track. As illustrated in Figure 3, while 6 out of 10 models exhibit rank differences of 3 or fewer, indicating general consistency, notable discrepancies also arise. For instance, Skywork-Reward-V2-Llama-3.1-8B performs substantially worse under PMDC than in the benchmark evaluation, suggesting that conventional metrics may overestimate its robustness in contentious, open-ended scenarios (see Appendix A2). These differences stem from PMDC's methodological distinction, where its discrepancy-driven sampling acts as a targeted probe, focusing on contentious cases with high inter-model disagreement. This approach emphasizes challenging edge cases and reduces sensitivity to potential train–test contamination, thereby offering a novel, complementary perspective relative to conventional benchmarks.

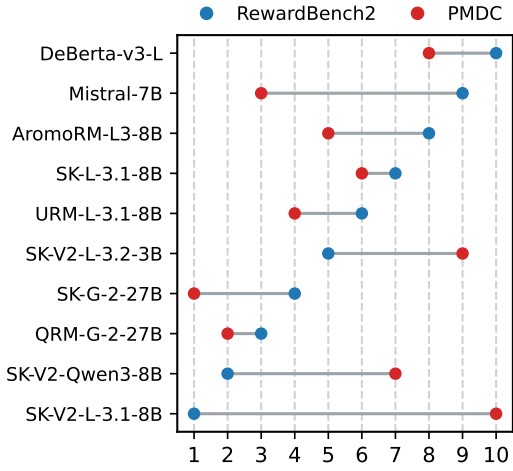

Figure 3: Rank comparison between PMDC and RewardBench2. Horizontal lines connect each reward model's ranking under RewardBench2 (blue) and PMDC (red).

Figure 4: PMDC's rank across 5 independent runs. The heatmap shows the rank of each RM in each run, with rank values annotated in individual cells.

### 4.3 MODEL AND RESULT ANALYSIS

**Sensitivity of Top-$k$**  To evaluate ranking stability, we vary the number of selected high-discrepancy pairs (top-$k$) from 2 to 100, using the ranking at $k = 100$ as the reference to compute Spearman correlation for each $k$. The results show that even with $k = 5$, the obtained rankings are highly consistent with those using $k = 100$, as reflected by Spearman correlation $0.988$ (see Figure 5). We also observe that very small $k$ values increase variance but help uncover edge-case disagreements, and rankings and metrics stabilize at $k \geq 5$. These findings support the use of a moderate $k$ to balance efficiency and robustness in practice.

**Annotation Efficiency Analysis**  The PMDC framework achieves remarkable annotation reduction compared to exhaustive evaluation. In our experiments, each of the $1,000$ questions contains 5 candidate responses, requiring $\binom{5}{2} = 10$ pairwise comparisons per question under traditional evaluation, resulting in $1,000 \times 10 = 10,000$ total annotations. In contrast, PMDC with $k = 10$ only requires $\binom{10}{2} \times 10 = 450$ comparisons, a $95.5\%$ reduction in annotation cost. This dramatic efficiency gain demonstrates PMDC's practical utility for large-scale reward model evaluation while preserving ranking fidelity.

**Sensitivity of LLM Judge**  To assess the robustness of PMDC against potential biases introduced by the choice of oracle, we evaluated the same set of Maximum Discrepancy samples using three distinct LLM judges: Claude-Sonnet-4, Gemini-2.5-Pro, and GLM-4-Plus. The resulting reward model rankings show remarkable consistency across all judge pairs: Claude vs. Gemini ($\rho = 0.964$), Claude vs. GLM ($\rho = 0.976$), and Gemini vs. GLM ($\rho = 0.976$). As visualized in Figure 6, the Top-3 models remain perfectly identical across all judges, with 5 out of 10 models showing complete ranking consensus, demonstrating that PMDC is insensitive to judge selection and captures consistent quality assessments across different oracle models.

**Result Consistency Analysis**  To assess the robustness of PMDC, we conducted five independent evaluation runs using different random seeds for prompt and LLM sampling. As illustrated in Figure 4, the global rankings remain highly stable across all runs, with most models maintaining identical or adjacent positions. In contrast, when using random sampling, i.e., selecting $k = 10$ pairs per model pair randomly from the same pool, the resulting rankings exhibit significantly higher variance across runs, as shown in Figure A1. This instability arises because random samples often fail to capture meaningful points of disagreement between reward models, leading to noisy and inconsistent comparisons. Instead, our PMDC approach produces reliable and reproducible evaluations.

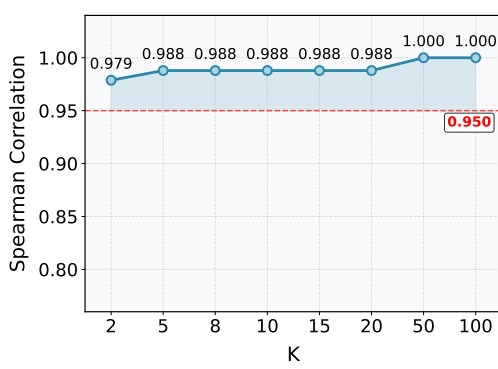
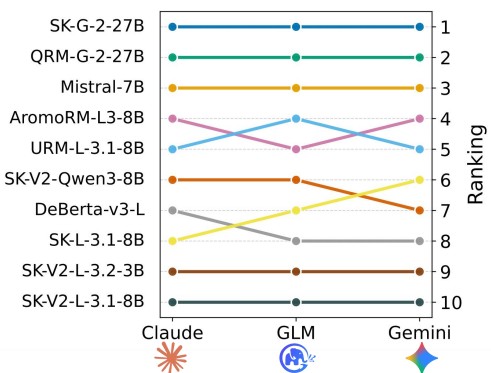

Figure 5: Spearman correlation of PMDC's ranks across top-$k$ values. The plot shows coefficients with a dashed red line at 0.95, highlighting all values above this threshold.

Figure 6: RM rankings across three Oracle judges (Claude-Sonnet-4, Gemini-2.5-Pro, GLM-4-Plus). The parallel coordinates plot shows each RM's rank across different judges.

### 4.4 IMPROVING REWARD MODELS VIA MAXIMUM DISCREPANCY PAIRS

Beyond evaluation, PMDC also can improve RMs through targeted fine-tuning. The maximum discrepancy samples identified by PMDC represent precisely those ambiguous or challenging cases where reward models exhibit substantial disagreement, making them ideal candidates for high-leverage fine-tuning. To validate this hypothesis, we train a reward model (baseline) based on ArmoRM-Llama3-8B-v0.1 (Wang et al., 2024) with the same implementation strategy, and fine-tune it using the 4500 oracle-annotated preference pairs selected by PMDC.

As shown in Table 2, fine-tuning on PMDC-selected samples yields an overall performance gain of 3.1% over the baseline on RewardBench2. The improvement is particularly notable in areas requiring nuanced judgment, such as *Math* and handling of *Ties*. Minor reductions are observed in *Focus* and *Safety*, likely due to the limited size and domain coverage of the fine-tuning set. Nevertheless, the overall improvement demonstrates that discrepancy-driven data selection can effectively boost reward model robustness and alignment fidelity. This result further validates the utility of PMDC not only as an evaluator but also as a data curation engine for reward modeling.

Table 2: Reward model performance before and after fine-tuning on PMDC-identified samples.

| Dimension | Baseline | Fine-tuned | Gain |
|-----------|----------|------------|------|
| Factuality | 0.534 | 0.545 | +1.1% |
| Focus | 0.642 | 0.588 | -5.4% |
| Math | 0.516 | 0.561 | +4.5% |
| Precise IF | 0.366 | 0.381 | +1.5% |
| Safety | 0.780 | 0.764 | -1.6% |
| Ties | 0.570 | 0.641 | +7.1% |
| **Overall** | **0.568** | **0.599** | **+3.1%** |

## 5 CONCLUSION AND FUTURE WORK

This work introduces the PMDC, a dynamic and efficient framework for evaluating RMs that overcomes limitations of conventional static benchmarks. By adaptively selecting high-discrepancy response pairs from a diverse and open-domain prompt pool and employing an LLM-based oracle for scalable preference judgment, PMDC enables robust, discrimination-rich evaluation of RMs. Empirical evaluation of 10 RMs not only reveals significant ranking inconsistencies compared to traditional benchmarks but also uncovers nuanced model-specific strengths and weaknesses. These results affirm PMDC's capacity to provide more generalizable, cost-effective, and behaviorally insightful assessments of reward model performance.

Future research could focus on enhancing oracle reliability through multi-judge ensembles, improving sample representativeness via adaptive sampling, and extending PMDC to multi-dimensional evaluation frameworks to better capture capability trade-offs. Scaling the framework to larger model sets and prompt pools would further enhance its robustness and applicability.

## ETHICS STATEMENT

This work focuses on evaluating reward models for aligning language models with human preferences. It does not involve the collection or analysis of personally identifiable, sensitive, or harmful data. All datasets used are publicly available benchmarks or synthetically generated using licensed models, with no private user content included. Human-like judgments were simulated via LLM-based oracles under carefully designed prompts to minimize bias and ensure consistency; no real human annotators were involved in this study. The proposed PMDC framework is intended solely for academic research and responsible model evaluation, with no foreseeable misuse toward generating, promoting, or optimizing harmful, deceptive, or unethical content.

## REPRODUCIBILITY STATEMENT

We are committed to ensuring the reproducibility of our work. Upon publication, we will publicly release the complete source code, and detailed configuration files for all experiments. Additionally, we provide the prompt template for the LLM-based oracle in the appendix. These resources will enable the research community to reproduce our results, validate our findings, and extend the PMDC framework for future reward model evaluation and development.

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

APPENDIX

## A1 ORACLE PROMPT

---

**Prompt for Oracle Judge Evaluation**

**System Message:** You are a professional text quality assessment expert. Please carefully compare the quality of two answers, focusing on: 1) Accuracy – whether the information is correct; 2) Relevance – whether it addresses the question; 3) Clarity – whether the expression is clear and understandable; 4) Conciseness – whether it is concise and avoids redundancy; 5) Depth – whether it has insights; 6) Logic – whether it is well-organized; 7) Practicality – whether it is helpful to the questioner. Find the best balance between information content and readability. Only return the result in JSON format, without any explanation.

**User Prompt:** Please judge which of the following two responses is better. Only return the result in JSON format without any explanation.

Question: {question}
Response 1: {response1}
Response 2: {response2}
Please answer strictly in the following JSON format: {"preference": 1} or {"preference": 2}
Where 1 means Response 1 is better, and 2 means Response 2 is better.

---

## A2 CASE STUDIES

The samples identified by PMDC naturally expose systematic evaluation divergences and contrasting assessment patterns across reward models. We present five representative cases from our empirical evaluation data that reveal how different reward models exhibit distinct preferences and evaluation criteria. These cases expose critical divergences in how models judge quality across scientific, creative, educational, and technical domains:

**Length Bias**: Some models prefer short answers over detailed ones, even when a deeper explanation is needed. (Cases 1, 4, 5);

**Inability to Recognize Nuanced Quality**: Some models fail to discern high quality in literary expression, complex reasoning, or specialized technical material (Cases 1, 2, 5);

**Context Insensitivity**: Some models use the same standard to judge all responses, even for different kinds of tasks that might need detailed analysis, creative expression, or technical expertise (Cases 1, 2, 3, 5);

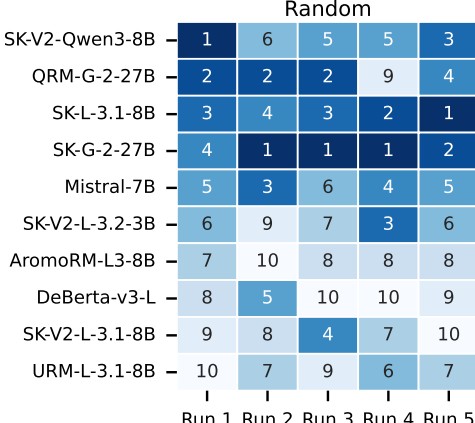

Figure A1: Random sampling's rank across 5 independent runs. The heatmap shows the rank of each RM in each run, with rank values annotated in individual cells.

These systematic evaluation patterns reveal fundamental differences in how reward models assess content quality, highlighting the importance of careful model selection for domain-specific applications and the need for more nuanced evaluation frameworks that can recognize diverse forms of excellence across diverse domains.

## A3 USE OF LARGE LANGUAGE MODELS

Large language models were used as assistive tools during the preparation of this manuscript. Specifically, they helped refine language, improve phrasing, and enhance overall readability. All LLM-generated suggestions were carefully reviewed, verified, and edited by the authors. No unverified

content was included, and all scientific reasoning, experimental design, analysis, and conclusions were solely the work of the authors.

**Case 1**: Skywork-V2-Llama-3.1-8B vs. ArmoRM-Llama3-8B-v0.1 comparison on scientific myth-busting. The former exhibits bias favoring brevity over comprehensive scientific analysis.

---

**Prompt:**

What will happen if one lemming jumps off a cliff?

---

| **Response A (Skywork Preference - Shorter):** | **Response B (Oracle Preference - Comprehensive):** |
|---|---|
| If one lemming jumps off a cliff, the outcome would depend on the specific circumstances, but here's what typically happens: | This is a fantastic question because it touches on one of the most persistent myths in the animal kingdom. The answer has two parts: the literal, physical reality and the debunking of the famous myth. |
| **1. In Reality**: Contrary to the popular myth, lemmings do not commit mass suicide by jumping off cliffs. This misconception was popularized by the 1958 Disney documentary *White Wilderness*, which staged the scene for dramatic effect. | **The Literal, Physical Answer**: If one lemming jumps off a cliff, it will fall. What happens next depends entirely on physics: If it's a short drop onto soft ground or into deep water, the lemming might survive. They are decent swimmers. If it's a high drop onto rocks or hard ground, the lemming will almost certainly be injured or killed by the impact. |
| **Migration & Accidents**: Lemmings may occasionally fall off cliffs or drown while migrating in large groups, especially when crossing unfamiliar terrain or bodies of water. However, this is accidental, not intentional. | Crucially, **nothing else will happen. The other lemmings will not mindlessly follow it over the edge**. |
| **2. If a Single Lemming Falls**: A lemming might survive a short drop if the cliff isn't too high and it lands safely (e.g., in snow or vegetation). However, a significant fall could injure or kill it. | **The Myth vs. The Reality**: The popular image of lemmings lining up to commit mass suicide by leaping off cliffs is completely false. This idea was created and popularized by a very specific source. |
| **No Group Effect**: Unlike the myth suggests, one lemming jumping would not trigger others to follow. Lemmings are not blindly obedient creatures. | **Where Did the Myth Come From?**: The myth was cemented in the public consciousness by the 1958 Disney nature film, "White Wilderness." The filmmakers brought lemmings to a cliff, used a spinning turntable to herd and force the frantic animals over the edge, and used clever editing to make it look like a willful, suicidal stampede. The scene was entirely staged. |
| **3. Ecological Impact**: A single lemming's death would have minimal impact on the population, as they reproduce quickly. However, predators (like owls or foxes) might scavenge the carcass. | **What Really Happens in Nature?**: Lemmings have dramatic population cycles. Every 3-4 years, their numbers explode, leading to mass migration to find new food and territory. During these migrations, accidental deaths occur when they try to cross obstacles like rivers and cliffs. These are accidental deaths, not intentional suicide. |
| **Key Takeaway**: The idea of lemmings mindlessly plunging to their deaths is a myth. In reality, they're simply small rodents that sometimes face hazards during migration—just like other animals. | |

---

**Oracle Judgment**: Response B (comprehensive myth-busting with detailed scientific explanation)
**Model Discrepancy**: Skywork-V2-Llama-3.1-8B (0.907 vs 0.319) strongly favored the shorter response, while ArmoRM (0.596 vs 0.745) and Oracle preferred the detailed scientific analysis.

---

**Response A's Limitations**: While factually correct, it provides only surface-level debunking without explaining the myth's origins or providing comprehensive scientific context. The explanation lacks depth in describing the actual biological behaviors of lemmings.
**Response B's Advantages**: Offers a structured, educational approach with clear separation between physical reality and myth debunking. Provides detailed historical context about the Disney film's staged footage, explains the actual population cycle mechanisms, and distinguishes between accidental deaths and intentional behavior. The response demonstrates superior scientific rigor and educational value.

---

**Case 2**: Skywork-Reward-Llama-3.1-8B vs. Skywork-Reward-V2-Llama-3.1-8B comparison on creative poetry evaluation. The latter exhibits bias in aesthetic preferences affecting creative content evaluation.

---

**Prompt:**
Write a funny, interesting, inspiring poem for Women's Day.

---

| **Response A (Oracle Preference - Modern & Bold):** | **Response B (V2 Preference - Traditional & Rhyming):** |
|---|---|
| **"Ode to the Glorious Glitch in the System"** | Here's to the woman, a creature of myth, |
| **Oh, Woman! You magnificent force of nature**, | Whose handbag contains what the universe smiths. |
| With your multitasking, your grace, and your *attitude*. | A charger, a lip balm, a singular sock, |
| You've changed the world with a laptop, a spatula, and a laser, | A half-eaten biscuit as hard as a rock, |
| While raising kids, running nations, and inventing the *attitude*. | A receipt from two-thousand-and-three, |
| **You were told to sit, so you stood.** | And the keys to a car that she sold. Who is she? |
| **Told to whisper, so you *roared*.** | She's a master of multitasking arts, |
| Told to stay in the kitchen— | Juggling deadlines and feelings and hearts. |
| So you turned it into a startup *and* baked a cake *and* ignored. | Her brain is a browser with sixty-four tabs, |
| **You've coded rockets, cured diseases**, | (Including the one tracking neighborhood cats). |
| Led revolutions in heels (and sometimes barefoot). | She carries the echo of those who once fought, |
| You've balanced budgets, babies, and boardroom decisions, | The suffragette's spirit that couldn't be bought. |
| All while being told, "Smile more, don't be so *ruthless*—wait, no, be *shrewd*?" | She can be a soft dawn or a hurricane's might, |
| **You are not "just" a mother, a wife, or a "lady."** | A comforting whisper or blaze burning bright. |
| **You are *also* a storm, a sonnet, a spark, and a scheme.** | She builds up an empire and builds up a friend, |
| You are Beyoncé on stage, Malala in class, | On her wisdom and strength, you can always depend. |
| And Aunt Linda who *still* won the neighborhood pickleball team. | So here's to her laughter, her power, her soul, |
| So here's to the women who *do*, who *dare*, who *demand*, | The one who makes broken things perfectly whole. |
| **Keep being weird, wise, wild, and wonderful.** | Happy Women's Day! Now go find your crown. |
| The world doesn't stand a chance. | (It's probably under a cushion somewhere in the lounge.) |
| *(And honestly? It never did.)* | |

---

**Oracle Judgment**: Response A (modern, bold voice with contemporary feminist themes)
**Model Discrepancy**: Skywork-Reward-Llama-3.1-8B (0.288 vs 0.096) favored Response A, while Skywork-Reward-V2-Llama-3.1-8B (0.275 vs 0.703) favored Response B, and Oracle preferred the bold creative approach.

---

**Response A's Advantages**: Feels more inspiring and empowering while maintaining humor. It takes more creative risks with its structure and language, creating a piece that's both entertaining and genuinely moving. The range of references (from Beyoncé to Malala to "Aunt Linda") makes it more inclusive and celebratory of all types of women. Demonstrates bold creativity and emotional impact.
**Response B's Limitations**: Beautifully crafted but plays it safer. While charming with traditional rhyme schemes and consistent meter, it doesn't reach the same emotional heights or creative boldness as Response A. The divergent model preferences reveal fundamentally different aesthetic philosophies in creative content evaluation.

**Case 3**: QRM-Gemma-2-27B vs. Skywork-Reward-V2-Llama-3.1-8B comparison on educational guidance assessment. The latter exhibits bias against comprehensive analytical responses.

---

**Prompt:**

Context: [header] How to stop an aggressive cat [title] Make a loud noise to distract the cat. [step] A noise can distract the cat and help redirect it away from its aggressive behavior. You could try clapping your hands, banging two objects, or ringing a bell.

Which ending makes the most sense?
A. [substeps] Making loud sounds to try to distract the cat can also distract it from other, more encouraging behavior. If the cat becomes aggressive, it may become defensive.
B. Direct the cat towards the noise to stop it and help it calm down. [substeps] Common loud noises to attract attention include a roaring vehicle or engine.
C. You could also use your voice, which might get the cat to come near you so that it more easily hears you. [substeps] Give the cat a low-pitched " ha-ha " sound.
D. [substeps] Cats can also be fearful of noises, so only use a loud noise if the cat is acting aggressively and you're afraid it could hurt someone, including another animal. If you use a loud noise to distract the cat, make sure it has a clear escape route in case it wants to flee.

---

| **Response A (QRM-Gemma Preference - Comprehensive Analysis):** | **Response B (Skywork Preference - Similar but with Style Elements):** |
|---|---|
| The most logical and complete ending is **D**. Here's why: | The most logical and contextually appropriate ending is: |
| **- D provides important context** about when and how to use loud noises to stop an aggressive cat. It warns that cats can be fearful of loud sounds and emphasizes that this method should only be used when necessary (if the cat is a danger to others). It also advises ensuring the cat has an escape route, which is crucial for preventing further stress or aggression. | **D. [substeps] Cats can also be fearful of noises...** **Reasoning:** |
| | - The original step recommends using a loud noise (like clapping or ringing a bell) to **distract** an aggressive cat. |
| **- A is contradictory** — it suggests loud noises might distract from "encouraging behavior" but doesn't clarify what that means, and it implies the cat may become defensive, which isn't helpful advice for stopping aggression. | - Option D provides a thoughtful, responsible follow-up: it acknowledges the potential fear response in cats, advises caution, and emphasizes safety for both people and the cat. |
| **- B is illogical** — directing a cat *toward* a loud noise would likely increase stress, not calm it down. The example of a "roaring vehicle or engine" is also impractical and overly extreme. | - It aligns well with humane behavior management by suggesting the noise be used only when necessary and ensuring the cat can retreat, reducing further stress or escalation. |
| **- C shifts focus to using the owner's voice**, which doesn't align with the original step (using loud noises to distract). The suggested "ha-ha" sound is unclear and not a standard cat behavior modification technique. | The other options are less appropriate: |
| | - **A** is confusing and contradictory — distracting from "encouraging behavior" doesn't fit the context. |
| **Thus, D is the best choice** because it offers practical, safety-conscious advice while staying relevant to the original step. | - **B** suggests directing the cat *toward* the noise, which is counterintuitive and potentially dangerous; also, a roaring engine is impractical and overly intense. |
| | - **C** introduces a strange "ha-ha" sound that doesn't align with standard cat behavior advice and shifts focus oddly to attracting the cat. |
| | Thus, **D** is the best choice. |

---

**Oracle Judgment**: Response B (clearer structure, better formatting, concise analysis)
**Model Discrepancy**: QRM-Gemma-2-27B (0.817 vs 0.883) favored Response A, while Skywork-Reward-V2-Llama-3.1-8B (0.760 vs 0.227) strongly favored Response A.

---

**Response A's Limitations**: While providing accurate systematic analysis and good breakdown of each option, it lacks the organizational clarity and precise language of Response B. The presentation is more dense and harder to scan, despite containing solid analytical content. The divergent model preferences reveal Skywork's systematic bias against well-formatted educational content, failing to recognize how superior presentation enhances learning effectiveness.
**Response B's Advantages**: Demonstrates superior organization with clear formatting, bold headers, and structured presentation that enhances readability. Uses more precise language ("contextually appropriate", "thoughtful, responsible follow-up") and explicitly mentions "humane behavior management" showing deeper understanding. The checkmark emoji and clean separation of reasoning sections make it easier to follow while maintaining analytical thoroughness.

**Case 4**: Skywork-Reward-Gemma-2-27B vs. Skywork-Reward-V2-Qwen3-8B comparison on healthcare AI applications. The latter exhibits bias against comprehensive technical content and practical implementation considerations.

---

**Prompt:**
Describe a scenario where artificial intelligence could be used to improve the quality and efficiency of healthcare delivery.

---

**Response A (Gemma Preference - Focused Narrative):**

Certainly! Here's a detailed scenario illustrating how artificial intelligence (AI) can enhance healthcare quality and efficiency:

**Scenario: AI-Powered Personalized Patient Management in a Hospital Setting**

**Context:** A large urban hospital aims to improve patient outcomes and optimize resource allocation in its cardiology department, which handles a high volume of patients with varying degrees of heart disease severity.

**How AI is Used:**

**1. Early Risk Detection and Diagnosis:** Upon patient admission, AI algorithms analyze electronic health records (EHR), including medical history, lab results, imaging scans (like echocardiograms), and even wearable device data. The AI model identifies patients at high risk of adverse cardiac events by spotting subtle patterns that might be missed by clinicians.

**2. Personalized Treatment Planning:** Based on the risk assessment, the system suggests personalized treatment plans. For instance, it recommends medication adjustments, lifestyle interventions, or scheduling for specific diagnostic tests tailored to the patient's unique profile.

**3. Real-Time Monitoring and Alerts:** Wearable devices monitor patients' vital signs continuously. AI detects anomalies such as arrhythmias or sudden changes in blood pressure and immediately alerts healthcare providers.

**Outcomes:** Improved Patient Outcomes, Increased Efficiency, Cost Savings, Enhanced Patient Experience.

**Response B (Oracle & V2-Qwen Preference - Comprehensive Structure):**

Artificial intelligence (AI) has the potential to significantly enhance healthcare delivery by improving quality, efficiency, and accessibility. Here's a detailed scenario illustrating how AI could be applied:

**Scenario: AI-Powered Early Disease Detection and Personalized Treatment in Primary Care**

**1. Patient Data Integration and Analysis** - A patient visits their primary care physician with vague symptoms like fatigue, weight loss, and occasional dizziness. The clinic uses an **AI-driven electronic health record (EHR) system** that aggregates data from past visits, lab results, wearable devices, and genetic information.

**2. Early Diagnosis and Risk Prediction** - The AI detects subtle irregularities and flags a **high risk for Type 2 diabetes and cardiovascular disease**, even though traditional diagnostic thresholds haven't yet been crossed.

**3. Personalized Treatment Planning** - The AI suggests **personalized lifestyle modifications** and **precision medicine interventions** based on the patient's genetic profile.

**4. Continuous Monitoring and Adaptive Care** - Real-time data feeds enable **automatic scheduling** and **prediction of potential complications**.

**5. Streamlining Administrative Tasks** - AI automates **appointment scheduling, insurance preauthorizations, and billing**.

**Benefits:** Early Detection, Reduced Costs, Personalized Care, Efficiency, Scalability.

**Challenges:** Data privacy, algorithmic bias, human oversight requirements.

---

**Oracle Judgment**: Response B (comprehensive structure, broader scope, practical considerations)
**Model Discrepancy**: Skywork-Reward-Gemma-2-27B (0.622 vs 0.912) strongly favored Response B, while Skywork-Reward-V2-Qwen3-8B (0.923 vs 0.383) strongly favored Response A with massive preference score discrepancy (0.830).

---

**Response A's Limitations**: While providing coherent cardiology-focused content, it demonstrates significant shortcomings in scope and depth. Offers generic references to "subtle patterns" and "anomalies" without concrete specificity. Lacks consideration of implementation challenges, administrative complexities, or ethical considerations. The narrow departmental focus misses broader healthcare system realities and presents an overly idealistic view without acknowledging practical barriers.
**Response B's Advantages**: Demonstrates superior content depth with comprehensive 5-stage coverage from data integration to administrative efficiency. Provides concrete specificity (actual symptoms, biomarkers, technologies) and excellent hierarchical organization with clear headers. Shows sophisticated understanding of healthcare complexities including privacy concerns, algorithmic bias, and human oversight requirements. Offers balanced perspective addressing both benefits and implementation challenges, making it more valuable for practical healthcare AI deployment.

**Case 5**: ArmoRM-Llama3-8B-v0.1 vs. Skywork-Reward-V2-Llama-3.2-3B comparison on physics education content. The latter exhibits bias against comprehensive educational formatting and visual learning aids.

---

**Prompt:**
What are the wavelengths of light emitted by metals depending on their temperature?

---

**Response A (Skywork Preference - Traditional Academic):**

The wavelengths (or spectrum) of light emitted by metals depending on their temperature are primarily described by **blackbody radiation** principles. When a metal is heated, it emits electromagnetic radiation across a range of wavelengths, and the distribution of this emitted radiation depends mainly on its temperature.

**Key Concepts:**

**1. Blackbody Radiation** - Metals, when heated enough, approximate a *blackbody radiator*—an idealized physical body that absorbs all incident electromagnetic radiation and re-radiates energy characteristic of its temperature.

**2. Planck's Law** - The spectral radiance $B(\lambda,T)$ of a blackbody at temperature T (in kelvins) as a function of wavelength $\lambda$ is given by Planck's radiation law: [Complex mathematical formula with h, c, $k_B$ constants]

**3. Wien's Displacement Law** - The wavelength $\lambda_{max}$ at which the radiation is most intense (peak wavelength) is inversely proportional to the temperature: $\lambda_{max} = b/T$ where $b \approx 2.898 \times 10^{-3}$ m·K is Wien's displacement constant.

**Typical Wavelengths at Different Temperatures:**
At ~500 K (227°C): Peak emission in infrared (around 5.8 $\mu$m), invisible to human eye. Around 1000 K (~727°C): Peak shifts to near-infrared (~2.9 $\mu$m), some thermal glow begins. At 1500 K (~1227°C): Peak near 1.9 $\mu$m, metal starts to glow dull red. At 2000 K (~1727°C): Peak around 1.45 $\mu$m, glowing brighter red-orange. At 3000 K (~2727°C): Peak shifts near 1 $\mu$m, metal glows bright orange-white. At 5000 K (~4727°C): Peak near 0.58 $\mu$m (yellow-green), very bright white light.

**Practical Notes:** Real metals do not emit as perfect blackbodies. Their emissivity (how efficiently they emit radiation) varies with wavelength and material. Thermal radiation spectrum is continuous, unlike atomic emission lines.

**Response B (Oracle & ArmoRM Preference - Interactive Educational):**

Of course! This is an excellent question that gets to the heart of a fundamental concept in physics called **black-body radiation**.

The short answer is: the light emitted by a metal is a **continuous spectrum of wavelengths**, and the **peak wavelength** depends almost entirely on its **temperature**, not the specific type of metal.

**The Key Principles** - Any object with a temperature above absolute zero emits thermal radiation...

**1. Wien's Displacement Law: The Color of the Glow** - **The Formula:** $\lambda_{peak} = b/T$. **What it means:** As temperature **increases**, the peak wavelength becomes **shorter**. This explains the color change: **Shorter Wavelengths** = Blue/Violet/UV, **Longer Wavelengths** = Red/Infrared.

**2. Stefan-Boltzmann Law: The Brightness of the Glow** - **The Formula:** $P = \varepsilon \cdot \sigma \cdot A \cdot T^4$. **What it means:** As temperature increases, brightness increases dramatically.

**Step-by-Step: Wavelength and Color vs. Temperature** - Let's walk through what happens as you heat a piece of iron:

**Comprehensive Temperature Table:** **~25°C**: **No visible light.** **~530°C**: **Faint, Dull Red Glow.** **~1000°C**: **Bright Cherry Red to Orange.** ... **~2500°C**: **"White Hot"** (tungsten filament temperature). **~5500°C**: **Bluish-White** (like the Sun).

**Visualizing the Spectrum** - [Includes blackbody radiation curve image] The **peak** shifts to shorter wavelengths as temperature rises, and **total brightness** increases massively.

**Important Distinction** - This continuous thermal radiation is **different** from atomic emission spectra. Metal incandescence produces a smooth, continuous spectrum determined by temperature.

---

**Oracle Judgment**: Response B (interactive educational approach, comprehensive table, visual aids)

**Model Discrepancy**: ArmoRM-Llama3-8B-v0.1 (0.021 vs 0.694) strongly favored Response B, while Skywork-Reward-V2-Llama-3.2-3B (0.389 vs 0.292) favored Response A with massive preference score discrepancy (0.770).

---

**Response A's Limitations**: Adopts overly theoretical approach starting with complex mathematical formulas that intimidate non-specialists. Information organization is scattered with practical applications buried in theoretical discussions. Temperature examples lack systematic progression and memorable associations.

**Response B's Advantages**: Demonstrates superior pedagogical design with clear "short answer" to detailed exploration progression. Features comprehensive temperature-color table with vivid descriptions ("Faint, Dull Red Glow," "White Hot") and practical anchors (tungsten filament, solar surface). Uses "What it means" explanations that bridge theory to intuitive understanding. Includes visual learning aids and distinguishes thermal from atomic spectra. Skywork's preference for Response A reveals systematic failure to recognize that effective science education requires both mathematical rigor and pedagogical accessibility.

