# OpenReview forum: "Dynamic Evaluation of Reward Models via Pairwise Maximum Discrepancy Competition"
_ICLR.cc/2026/Conference — ICLR 2026 Conference Withdrawn Submission_

### Official Review · Reviewer_FdQs · 2025-10-23

**Soundness:** 2
**Presentation:** 2
**Contribution:** 2
**Rating:** 2
**Confidence:** 4

**Summary:**

This paper proposes PMDC, A dynamic, annotation-efficient evaluation framework that selects “maximum discrepancy” prompt-response pairs using multiple reward models. An RM evaluation  dataset via active probing is constructed to identify highly divergent reward model responses. Experiments demonstrate that it not only enables discrimination-rich evaluation of reward models but also enhances downstream alignment performance.

**Strengths:**

1. Targeted evaluation via discrepancy sampling: The selection objective explicitly maximizes |(sA(q,a1)−sA(q,a2)) − (sB(q,a1)−sB(q,a2))|, which focuses the RM evaluation efforts on the most contentious cases.

2. Clear and implementable benchmark construction pipeline: The framework specifies the data path end-to-end: prompt pooling, response sampling, per-RM scoring, top-k discrepancy selection per RM pair, LLM-judge 2AFC, and Bradley–Terry modeling.

**Weaknesses:**

1. The main contribution of this work is that it demonstrates a significant ranking inconsistencies of reward models between PMDC and  traditional benchmarks. These inconsistencies stem from the difference in benchmark data construction: PMDC focuses on the most contentious cases, while traditional benchmarks do not. This bias seems obvious and intuitive, making the contribution of this work unclear.  What are the useful insights that can be derived or concluded from this benchmark?

2. There exists inconsistency in normalization vs discrepancy computation. Eq. 1 introduces min–max normalization, but Eq. 3/4 use s_A,s_B without primes. If discrepancies are computed on unnormalized scores, scale differences between RMs can distort sample selection; if they are normalized, the notation and implementation details need to be corrected and made explicit.

3. Bradley–Terry likelihood specification (Eq. 7) appears incorrect: It shows [wij log Pi,j + wji log Pi,j], but the second term should be wji log Pj,i.

4. 2AFC protocol conflicts with “oracle can’t decide” narrative: Section 3.3 mentions “oracle cannot determine a superior response,” yet PMDC uses a strict 2AFC oracle prompt with no tie.

5. Sensitivity to extreme scores and sampling composition: Min–max normalization across a large, mixed prompt set is highly sensitive to outliers, impacting both preference discretization and discrepancy magnitudes.

6. The fine-tuning setup of Section 4.5 lacks details such as training data construction and training details.

**Questions:**

1. Compatibility of the evaluated reward models (Table 1) and datasets. What in specific do these reward models optimize, helpfulness, honesty or others? Are these objectives compatible with the evaluation datasets?

2. In Equation (8), Agreement(Ri) is measured only on PMDC-selected contentious cases. Can you report agreement on a matched random subset to quantify how much your metric reflects “performance on controversy” vs overall accuracy?

---

### Official Review · Reviewer_d9BR · 2025-10-25

**Soundness:** 2
**Presentation:** 3
**Contribution:** 2
**Rating:** 2
**Confidence:** 3

**Summary:**

The submission is in the "datasets and benchmarks" area, which proposes Pairwise Maximum Discrepancy Competition (PMDC) — a dynamic and annotation-efficient framework for evaluating RMs. Instead of evaluating each RM on a fixed dataset (e.g., RewardBench 1/2), PMDC adaptively identifies prompt–response pairs where two RMs most disagree, then asks a strong LLM to decide which RM’s preferences align better with "human" judgment. The method aggregates these pairwise comparisons to derive global rankings of RMs.

Experiments re-evaluate ten public RMs using prompts aggregated from multiple sources (MMLU, GSM8K, HumanEval, TruthfulQA, etc.), finding large deviations from RewardBench2 rankings. The authors claim PMDC offers better generalization assessment and significant annotation-cost savings. They also show small downstream gains (+3.1%) when fine-tuning an RM on PMDC-identified high-discrepancy samples.

**Strengths:**

There are several strengths of this submission:

1. The dynamic-evaluation framing is new to the reviewer's knowledge. PMDC adapts the “Maximum Discrepancy Competition” ideas to RM evaluation, emphasizing active probing over static test sets.

2. The arguments around efficiency and automation of the annotation process is valid. Using an LLM-as-judge oracle avoids costly human annotation and identifies the most discriminative cases, though clearly with caveats listed below.

3. The study compares ten prominent RMs from multiple families, reporting both ranking and oracle-agreement metrics, and analyzes stability across seeds, judges, and top-k values. This setup is good.

4. The paper also demonstrates that PMDC-selected samples can improve RMs through targeted fine-tuning, which is interesting.

**Weaknesses:**

However, though with several novel points, the reviewer is not convinced that the proposed evaluation framework is sufficiently sound and will be preferred over existing evaluation protocols, due to the following weaknesses:

1. Dependence among RMs and limited practical scalability. PMDC compares RMs pairwise and selects “maximum discrepancy” prompts jointly over all N(N−1)/2 pairs. This inherently introduces dependency among models — unlike existing evaluations that independently score each RM. Such dependency limits the method’s use for leaderboards or community evaluation due to clear scalability issues.

2. The reliance on an “advanced LLM oracle” as ground truth clearly has caveats. The paper reports high cross-judge Spearman correlations but lacks convincing human validation to confirm that any oracle is trustworthy.

3. Because each RM pair may be evaluated on different prompt subsets, PMDC does not ensure that all models are assessed on the same samples. The resulting ranking may have unknown statistical variance or dependence on sampling randomness. No rigorous or convincing analysis is provided on this front.

4. High probability of potential leakage. It seems that the paper focuses on more the "methodology", but the exact setup is critically important for the topic concerned, which is unfortunately not very rigorous to the reviewer. Specifically, the “open-domain prompt pool” is built by aggregating well-known datasets such as MMLU, GSM8K, TruthfulQA, etc. all potentially heavily used in pre-training or alignment of the RMs being evaluated.

5. Limited discussion of how the community can use PMDC. This is important for a Dataset & Evaluation-track paper. Without a public pipeline or easy-to-extend design, its community impact may be small.

**Questions:**

Please refer to the detailed reviews. Also:

Have you performed any human-oracle agreement studies to quantify the accuracy of the LLM judge compared to real human preferences?

Can PMDC support incremental evaluation (the typical scenario when a researcher has one new RM)?

Can you provide confidence intervals for the Bradley–Terry scores or conduct bootstrap analysis to show ranking robustness?

---

### Official Review · Reviewer_9evW · 2025-10-26

**Soundness:** 2
**Presentation:** 3
**Contribution:** 2
**Rating:** 4
**Confidence:** 3

**Summary:**

The paper introduces PMDC as a dynamic, annotation-efficient way to evaluate reward models by actively sampling the the data pairs eliciting maximum discrepancy across the two reward models under comparison, judging them with a strong oracle, and aggregating pairwise comparison using a Bradley-Terry model. On 10 representative RMs, PMDC method reveals a different hierarchy than static benchmarks RewardBench2. Finetuning an RM on PMDC-selected, oracle-labeled pairs improves RewardBench2 overall score by +3.1%, especially on math and tie-handling. The paper provides a novel, competition-based reward model benchmarking, different from traditional static benchmarks.

**Strengths:**

Originality: I think this competition based ranking of reward model is quite novel and I like the idea of sampling maximum discrepancy data for judging, largely reducing the efforts of data labeling.
Quality: The paper does try to provide different ablation studies like testing result robustness against different oracle judges, or different prompt/responses creation.
Clarity: The results for establishing the new evaluation method are clearly presented and the case studies examples in the Appendix are helpful for understanding.
Significance: The method offers a potential alternative to static benchmarks if the authors can provide more details on how to scale up the practical usage.

**Weaknesses:**

1. Section 4.2 and Figure 3: I think the claim that "6 out of 10 models exhibit rank differences of 3 or fewer, indicating general consistency" is not rigorous. RewardBench2 is a non competition based benchmark so its ranking is absolute ranking. Besides these top 10 models there are many other models. In the experiment, authors only limit to pairwise comparison within the 10 models so there are only 10 rankings these models can take. Hypothetically if using this method to evaluate the whole pool, the ranking might not be limited to that difference.
2. Point 1 leads to another potential weakness. The method is motivated partly such that the evaluation is more data efficient (only picking the high discrepency data points). However, when ranking a large pool of reward models (like the numbers in rewardbench 2), pairwise comparison will end up be more costly. This limits the impact and practical usefulness of the method
3. I think Experiment 4.4's logic is a bit problematic. The authors discusses that using these maximum discrepancy pair data for finetuning reward model can improve the performance and the evaluation criteria is rewardbench2. However, in earlier part Figure 3, the authors just demonstrate that sometimes their evaluation method can be quite different from rewardbench2 and suggest that this is a potential weakness of traditional benchmark. I am confused by what should be a gold standard here.

**Questions:**

1. Regarding weakness 1 mentioned, do authors have other methods to compare the result with the standard benchmarks?
2. Regarding weakness 2 mentioned, do authors have any methods to generalize this method to large pool reward model comparison?
3. Regarding weakness 3 mentioned, can authors provide better logic on how they view the difference of their evaluation vs traditional benchmarking methods? if they think theirs is superior, are there other methods to illustrate the performance gain of training on maximum discrepancy pairs data (as rewardbench is no longer reliable)? if the authors think rewardbench is still valid/gold standard, please provide more reasoning on the contribution of this evaluation method?

---

### Official Review · Reviewer_AE36 · 2025-10-29

**Soundness:** 2
**Presentation:** 3
**Contribution:** 2
**Rating:** 2
**Confidence:** 3

**Summary:**

The paper proposes Pairwise Maximum Discrepancy Competition (PMDC), a framework for evaluating reward models (RMs) by identifying prompts that cause large disagreements among models and resolving them with an LLM-based oracle judge. The judgments are aggregated using a Bradley–Terry (BT) model to derive a global ranking. Experiments on ten reward models from RewardBench show different rankings than static benchmarks, and fine-tuning on PMDC-selected samples slightly improves downstream performance.

**Strengths:**

1.	Clear formulation and structure. The PMDC pipeline and selection rule are mathematically consistent and easy to follow.
2.	Dynamic testing idea. The notion of adaptively probing contentious prompts is intuitive and could help uncover edge cases that static benchmarks miss.
3.	Readable and reproducible. The authors describe implementation details, oracle prompts, and ablations clearly.

**Weaknesses:**

1.	Measures relative ranking, not true accuracy. The framework produces pairwise rankings rather than evaluating how well each RM aligns with human preferences in absolute terms. Without a human reference or calibrated oracle, it cannot measure quality—only which model “wins” more often.
2.	Heavy dependence on LLM oracles. The oracle’s biases and inconsistencies directly shape the final ranking, yet no human or inter-oracle validation is presented to confirm reliability.
3.	Scalability issues. The Bradley–Terry model requires O(N^2) comparisons for N reward models, which becomes infeasible for large collections (e.g., hundreds of RMs). Sparse comparisons lead to unstable or ill-defined rankings.
4.	Limited empirical depth. Experiments cover only ten RMs and one oracle; no statistical error bars, human correlation checks, or domain diversity analyses are provided.
5.	Incremental conceptual value. The method mostly repackages known maximum-discrepancy and BT ideas, with limited novelty in algorithmic insight or theoretical grounding.

**Questions:**

1.	How can PMDC’s ranking be converted into an absolute evaluation of each RM’s correctness or calibration, rather than a relative order?
2.	Have you verified that the LLM oracle’s judgments correlate with human preference annotations?
3.	How would PMDC scale to large RM sets (e.g., 100–300 models)? Is there a way to approximate the BT rankings without quadratic comparisons?

---

### Note · Authors · 2025-11-23

I have read and agree with the venue's withdrawal policy on behalf of myself and my co-authors.